# Drinking Water Investigation of Hill Tribes: A Case Study in Northern Thailand

**DOI:** 10.3390/ijerph17051698

**Published:** 2020-03-05

**Authors:** Suntorn Sudsandee, Krailak Fakkaew, Vivat Keawdounglek, Pussadee Laor, Suwalee Worakhunpiset, Tawatchai Apidechkul

**Affiliations:** 1Environmental Health Program, School of Health Science, Mae Fah Luang University, Chiang Rai 57100, Thailand; krailak.fak@mfu.ac.th (K.F.); vivat.kea@mfu.ac.th (V.K.); pussadee.lao@mfu.ac.th (P.L.);; 2Department of Social and Environmental Medicine, Faculty of Tropical Medicine, Mahidol University, Bangkok 10400, Thailand; suwalee.wor@mahidol.ac.th; 3Center of Excellence for the Hill-tribe Health Research, Mae Fah Luang University, Chiang Rai 57100, Thailand

**Keywords:** hill tribes, mountain water supply, drinking water, turbidity, biological parameters

## Abstract

Hill tribes are a group of people who live in remote areas in northern Thailand. They typically use untreated water for drinking, that can lead several health problems. The six main hill tribes—Akha, Hmong, Karen, Lahu, Lisu, and Yao—were selected for the study. A validated questionnaire was used for data collection. Water samples were collected from the selected villages and tested for the quality at Mae Fah Luang University, Thailand. Results: the major sources of drinking water were mountain water supplies (74.3%), and commercial bottled water (21.4%). Easy access, sufficiency for the whole year, and food-drug administration sign labeled were the criteria used for selecting sources of drinking water. Colorless and safety were also used as a selection criteria for their drinking water in some tribes. Lisu, Karen, and Hmong treated their drinking water by boiling, while Akha and Lahu stored the water in certain containers to allow particle settling before drinking water without treatment. 42.0% of the water samples had a turbidity values <5 NTU, and total coliform and fecal coliform bacteria were detected in 100.0% of the samples. To prevent water-borne diseases among the hill tribe people, appropriate water treatments such as boiling, filtration, and disinfection are recommended.

## 1. Introduction

Adequate and safe water supply, especially drinking water, must be available for all people, which can result in a benefit to health and boost countries’ economic growth as well as contribute greatly to poverty reduction. In 2017, about 5.3 billion people had access to safely managed drinking-water services, but the remaining 2.2 billion people still do not have that access. The United Nations (UN) and World Health Organization (WHO) found that more than 40.0% of the global populations did not have access to sufficient clean water, and the WHO estimated that 144 million people are collecting and using untreated surface water from lakes, ponds, rivers and streams. Specifically, people who live in mountainous area, especially in low-income countries, informal, or illegal settlements usually have less access to safe drinking water than other residents [1,2,3]. Globally, the lack of access to safe drinking water and sanitation accounted for 88.0% of all deaths from diarrheal disease, especially in children under five years [4,5]. Drinking water contamination originates from human and animal wastes from defecation close to water sources [6], cross-contamination from sewage lines [7], sewage disposal without treatment and leakage from septic tanks or pit latrines [4]. Globally, at least 2 billion people used drinking water sources contaminated with feces [2]. In Thailand, drinking water contamination by microorganisms has been recognized as a major problem. Several microorganism contamination consequences have been reported such as severe diarrhea, dysentery, hepatitis A, and intestinal parasitic infections [8,9,10,11].

Improving access to safe drinking water can eventually benefit human health by reducing the risk of water-borne infectious diseases [3,12,13]. In Thailand, the Provincial Waterworks Authority (PWA) and the Metropolitan Waterworks Authority (MWA) are the major organizations that produce and distribute the water supply to households. The general water treatment system begins with raw water undergoing coagulation and flocculation, sand filtration, and disinfection. After these processes, the treated water is then distributed to the consumers [14,15,16]. However, the PWA and the MWA services do not cover the whole area of Thailand. They manage and source clean water only to the people living in the big cities and municipal areas, while, people in rural areas, especially in hill tribe villages, are out of the service zones of both the PWA and the MWA.

Most hill tribe villages are located in the mountainous areas in Northern Thailand. The hill tribes in Thailand are divided into six main groups: Akha, Hmong, Karen, Lahu, Lisu, and Yao [17]. They have migrated from the southern part of China into Thailand in recent decades. Since 2012 approximately 800,000 hill tribe people live in Thailand [18]. Each tribe group has their own language, culture, and beliefs, which are different from those the Thai people. Most hill tribes live in the mountainous border areas of the 16 northern provinces in Thailand. In 2012, 180,214 hill tribe members lived in 652 villages in Chiang Rai Province [17] where elevations range from 405–1300 m above sea level (MASL). The villages are generally outside the service areas of the PWA. This study aims to investigate the system of drinking water management, and detect its quality, including assessing people knowledge and practices relevant to their daily drinking water consumption.

## 2. Materials and Methods

### 2.1. Study Sites

Six major hill tribes; Akha, Hmong, Karen, Lahu, Lisu, and Yao, living in mountain areas of Chiang Rai Province, northern Thailand, were selected. The study sites are shown in Figure 1. The Karen and Lahu villages are located the center of Chiang Rai Province, while the Akha, Yao and Lisu villages are located along the Thailand-Myanmar border. The Hmong villages are located along Thailand-Laos border. All study sites were determined on the 72 locations and elevated by using global position system (GPS) (Garmin Oregon 750, Metro Area, KS, USA).

### 2.2. Household Survey and Water Sampling

This study was approved by Human Ethics Committee of Mae Fah Luang University (No. REH-62101). Household survey was conducted for gathering data on socioeconomic data and practices on drinking water management. Questionnaires were adopted from literature review, primary community survey and experts’ consultations. The content validity was improved by using the item-objective congruence (IOC) method [19]. The Cronbach’s alpha was then analyzed to confirm the reliability of this questionnaire which was found at 0.87. Two types of questionnaire (water drinking behavior and drinking water management) were employed to gather information from the participants. Participants were randomly selected from 40.0–100.0% of the households who lived in selected villages.

The variables in the study were sourced from the literature review. Validated questions were used to acquire information relevant to drinking water selection of the respondents, based on a five-point Likert scale; from 1 (strongly disagree) to 5 (strongly agree). While the village leaders were asked to obtain the drinking water management in their village. The data obtained from questionnaires were further analyzed.

Water samples were collected by grab sampling from different sources; (1) water sources on the mountain, (2) water storage tanks, and (3) water at households, between March–April 2019, using polyethylene and glass bottles chilled at 4 °C in an ice box. A total of 24 water samples were analyzed for physical and microbiological characteristics at Environmental Health Laboratory, Mae Fah Luang University. Turbidity of the water samples was measured by a turbidity meter (Turb 43 IE, WTW Wissenschaftlich, Weilheim, Germany). The biological tests, total coliform bacteria and fecal coliform bacteria were analyzed by the 9221 B and 9221 E standard methods [20].

### 2.3. Statistical Analysis

A total of 425 questionnaires obtained from the six hill tribe groups were organized for statistical analysis using the SPSS version 22.0 software (IBM Company, Armonk, NY, USA). Normality tests were used Kolmogorov-Smirnov test. One-way ANOVA was applied to compare the mean scores among sample groups, whereas *t*-test for parametric data and Kruskal-Wallis test were applied to compare the mean scores among sample groups for non-parametric data. The significant difference was considered if the *p*-value was less than 0.05.

## 3. Results and Discussions

### 3.1. The Participnats’ Profiles

A total of 2115 persons of 425 households were recruited into the study. The participants’ characteristics are summarized in Table 1. More than half of the participants were female (55.3%), and the order of participants’ educational level was primary or below > secondary > bachelor degrees. Most of the participants worked in the agricultural sector (62.6%) such as corn farms, green tea farms, and rubber tree farms, followed by freelance (25.9%), and their monthly incomes were less than 10,000 baht. Moreover, they did not have access to primary education, especially those who were stateless with no Thai citizenship.

### 3.2. Sources of Drinking Water

Most of the participants (six hill tribes) obtained their drinking water from a mountain water supply (74.95%), followed by commercial bottled drinking water (21.05%), shallow wells (3.29%), groundwater (0.47%), canal water (0.24%) and other sources (0.24%) such as rain water, spring, and small ponds. In summary, the sources of drinking water of each hill tribe group are shown in Figure 2. The Akha and Lahu people used two sources of drinking water, with more mountain water supply being used than commercial bottled drinking water. For the Hmong people, three sources of drinking water were used: a mountain water supply and two other sources: groundwater and canal. The Karen people consumed drinking water from four sources: mountain water supply, commercial bottled drinking water, shallow wells, and groundwater. The Lisu people used two sources of drinking water: mountain water supply and shallow wells. Meanwhile, the Yao people used three sources of drinking water: mountain water supply, commercial bottled drinking water, and shallow wells.

The mountain water supply was the major source of drinking water source among the hill tribes. This result is similar to a previous study which reported that mountain water was a primary source of freshwater supplied in hilly regions inhabited by the hill tribe people in a mountainous area of Thailand [9,11,21]. Commercial bottled drinking water was also a source of drinking water for the Yao, Karen, Akha, and Lahu people, but not for Hmong and Lisu people. Some people believe that commercial bottled drinking water is more convenient and better tasting than tap water, therefore, it has been widely used by people worldwide [22]. However, the Hmong and Lisu do not prefer commercial bottled drinking water. This might be the difficulty of transportation by a company to deliver it to their villages. Besides, it was found that some hill tribe people used shallow wells water, particularly among the Lisu, Hmong, and Karen people for whom it accounted for 12.2%, 7.5%, and 1.3% of the total consumption, respectively. The less favor of shallow wells water of the hill tribe people could be due to the accessibility of a larger source of water in their villages.

### 3.3. Influencing Factors for Some Selcted Sources of Drinking Water of the Hill Tribe People

The criteria of “colorless or clear” were used for selection of their sources of drinking water among the Akha, Hmong, and Karen people. Colorless or transparent water can be measured as its turbidity. Water with turbidity value less than 5 NTU is defined as clear water, and an individual can begin to see cloudiness with the naked eyes when the turbidity value is greater than 5 NTU [12,23,24]. Meanwhile, the criterion of “health safety” was the greatest concern for selection of their sources of drinking water of the Lahu, and Lisu people. This concern would relate to the impact to their health after drinking water particularly had experienced of water-borne diseases such as diarrhea, and dysentery.

The Yao people chose “no odor” as their major concern for selection their drinking water. Odor can be generated from algae or microorganisms in a water source, resulting from growth, activities or metabolism [25]. Colorless, taste, and odor are used for selection the drinking water of a wide number of consumers [12]. Regarding the source of drinking water, it was found that the Yao people selected a mountain water supply about 55.3%, which was a lesser proportion than other tribes, but they chose the commercial bottled drinking water with greater proportion than other tribes.

According to the participants’ opinion on the criteria on selection their drinking water three major criteria; colorless or clear, health safety, and no odor. There were different opinions in selection criteria in different tribes (Table 2).

The “easy access”, “sufficiency the whole year” and “FDA standard” were used as the determination criteria in the selection of sources of drinking water which were available in three sources; mountain water supply, commercial bottled drinking water, and shallow wells; the results are shown in Figure 3.

It was found that in the criteria of “easy access”, those who used the mountain water supply, and shallow wells were significantly greater than those bottled drinking waters (*p*-value < 0.05). Considering “sufficiency the whole year”, the mean score of those who used shallow wells was significantly greater than those used commercial bottled drinking water (*p*-value < 0.05). In addition, for the “FDA standard” criteria, the mean score of those who used commercial bottled drinking water was significantly greater than those who used the mountain piped water supply (*p*-value < 0.05) (Table 3).

Due to the hill tribe people live in the mountainous areas, the mountain water supply is a primary source of their freshwater [26], whereas, few people used commercial bottled drinking water because the difficulty of access, although some tribes like the Yao (45.3%), Karen (33.7%), Akha (24.1%), and Luha (13.9%) favored the use of commercial bottled drinking water. These groups were more concerned about the FDA standards for selection their sources of drinking water. The knowledge about safe drinking water was significantly high among the Yao people compared to the others (*p*-value < 0.05).

### 3.4. Mountain Water Supply Management

The mountain water supply management of the six hill tribes is summarized in Table 4 and Figure 4. Mountain stream water is obtained from 1–7 raw water source points, depending on the topography of the village location of each tribe. The concrete dams, soil dams and water storage concrete-tanks were used for storing the mountain stream water where were far from their villages, about 1–6 km. The water was supplied through 2–3 inch PVC pipes that distribute the water to storage tanks in their villages. The water was stored in these tanks to allow suspended particles to settle out of the water before further distribution to households using 1½ inch and ½ inch PVC pipes. This mountain water supply system is commonly applied in hilly regions [21].

The maintenance cost for this system was approximately 25–151 US dollar/month, depending on the frequency of water pipeline repairs. There were 2–3 administrators who were selected from the village members and responsible for the system management and maintenance. There were several problems that required management; (1) solid particles accumulated at the bottom of the dams and tanks which need to be cleaned 2–3 times per year, (2) PVC pipe damage, (3) raw water pollution from nearby communities, (4) turbidity increases during rainy seasons and (5) inadequate raw water availability during the dry season. Under certain condition, the village leaders preferred several points: (1) each household should install a water filter unit, (2) disinfection units (chlorine, UV, or ozone) should be installed on the central water supply of each village to treat water before distributing it to households, and (3) larger tanks should be built to improve the sedimentation efficiency.

### 3.5. Houshold Water Storage Containers

The hill tribe people have adopted different types of containers to store the water for daily use and drinking. Based on the convenience, plastic tanks or bottles were the greatest favorite water container for the Hmong, Lisu, and Yao people, while some households of the Hmong and Yao used a traditional jar. The Karen village is the most famous tourist destination, and large concrete tanks were used for storage water. Some households of the hill tribes used the water directly from water taps without storage containers.

### 3.6. Post-Treatment of Drinking Water

The results of drinking water treatment at a household level of the six hill tribes are shown in Figure 5.

Boiling is the most favorite method for water treatment. This method is an effective method for drinking water treatment at the household level and widely accepted by health professionals, especially in developing countries [27,28]. The Lisu (92.0%), Karen (74.0%), and Hmong (72.0%) people boiled the water before drinking, respectively. Meanwhile, most of the Akha and Lahu people favored storing the water in containers to allow particle settling before drinking without treatments. These results were similar to the previous reports [9,29]. This might be due to the belief of some tribes that boiled water is given only to an ill person [30]. About 40.0% of the Yao people used a water filter for treating their drinking water. A study recommended that to improve the quality of drinking water among the hill tribe people the use of a filter could be effective [31].

### 3.7. Drinking Water Quality

The turbidity and biological parameters of the selected water samples are shown in Table 5. It was found that the turbidity of all samples from the Hmong and Karen villages met the drinking water standards [12,32], while the turbidity of the water samples from the Lahu villages were over the standards. This might be caused by having a small storage tank at the villages that was not suitable for suspended particle sedimentation [33]. To improve the sedimentation efficiency, the size of storage tanks should be designed based on the water velocity and its volume [33], especially in the rainy season, when small particles, sediment, suspended solids, bacteria, and organic substances from storm water run-off are major causes of the high turbidity of the raw water [21]. 

Regarding the biological parameters, it was shown that total coliform bacteria and fecal coliform bacteria were detected in all samples. This is similar to several previous studies on the biological contamination in drinking water from mountain water sources [9,34,35,36]. This reflects that all the hill tribe waters were not good for drinking because they did not meet the drinking water standards [12,32]. A simple disinfection method, boiling, which can inactivate of about 99.0% of geometric mean fecal coliforms [37], is recommended.

## 4. Conclusions

The hill tribe people are the group of people who live in the high mountains, and in the remote areas where they live it is definitely difficult to access clean drinking water. Most of the water supply and drinking water are obtained from untreated natural sources of water such as mountain water supplies, groundwater and shallow wells. The “colorless or clear” and “health safety” criteria are the major concerns for their selection of their drinking water. Moreover, the “easy to access”, “sufficiency for the whole year”, and “having FDA label” are influencing factors for the selection their sources of drinking water. The hill tribe people construct simple systems of mountain water supply without a proper treatment to use in their households, and 42.0% of the hill tribe drinking water has a turbidity value over 5 NTU, and 100.0% are contaminated by coliform bacteria and fecal coliform bacteria. Public health interventions should be implemented to promote having clean drinking water among the hill tribe people in Thailand by treating it properly.

## Figures and Tables

**Figure 1 ijerph-17-01698-f001:**
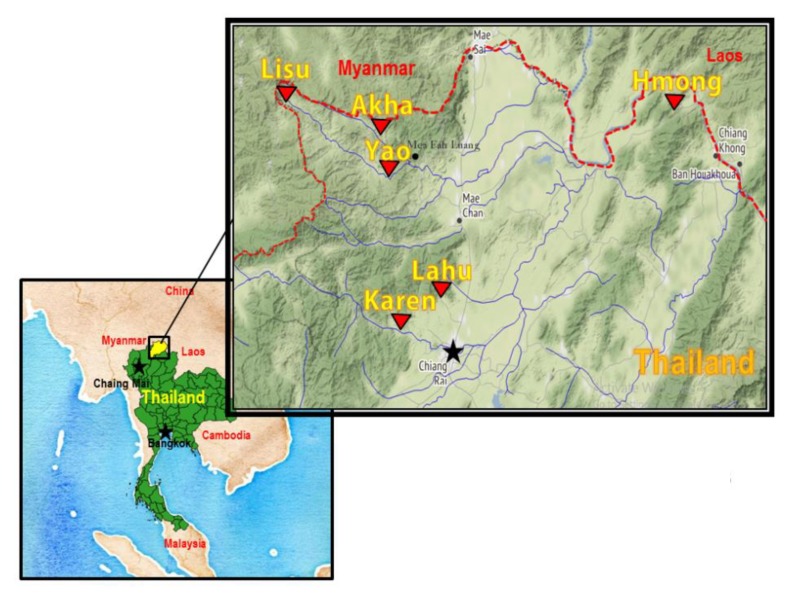
Study sites of six hill tribes in Chiang Rai province, Thailand.

**Figure 2 ijerph-17-01698-f002:**
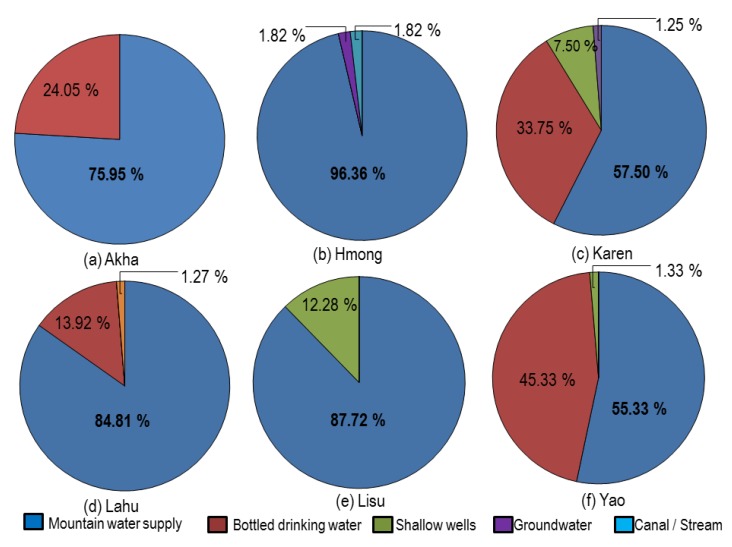
(**a–f**): Drinking water sources of six hill tribe groups.

**Figure 3 ijerph-17-01698-f003:**
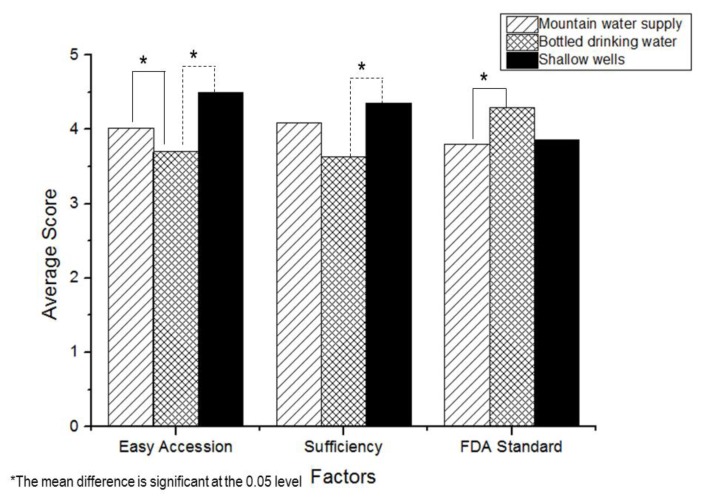
Comparison of tree influencing factors of drinking water.

**Figure 4 ijerph-17-01698-f004:**
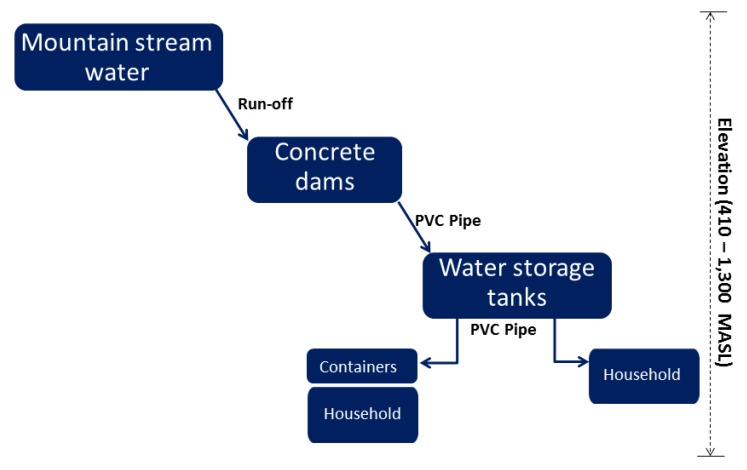
Mountain water supply management of hill tribe groups.

**Figure 5 ijerph-17-01698-f005:**
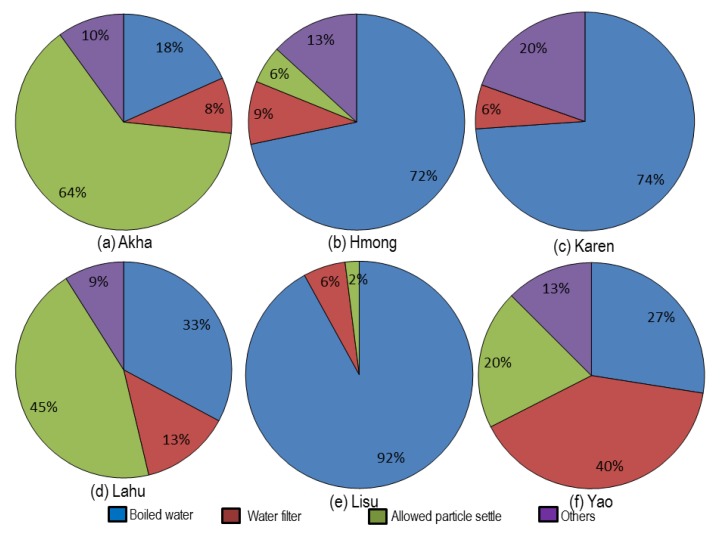
(**a**–**f**): Drinking water treatment at household level.

**Table 1 ijerph-17-01698-t001:** Sociodemographic characteristics of respondent in six hill tribe groups.

No	Variable	Total Number of Respondent (*n* = 425) (%)	Number of Respondent (% of Each Tribe)					
			Akha (*n* = 79)	Hmong (*n* = 55)	Karen (*n* = 80)	Lahu (*n* = 79)	Lisu (*n* = 57)	Yao (*n* = 75)
1	Gender							
	Male	190 (44.7 %)	34 (43.0%)	41 (74.5%)	33 (41.3%)	33 (41.8%)	24 (42.1%)	25 (33.3%)
	Female	235 (55.3%)	45 (57.0%)	14 (25.5%)	47 (58.8%)	46 (58.2%)	33 (57.9%)	50 (66.7%)
2	Education							
	Primary or below	329 (77.4%)	69 (87.3%)	38 (69.1%)	50 (62.5%)	58 (73.4%)	54 (94.7%)	60 (80.0%)
	Secondary and Tertiary	86 (20.2%)	10 (12.7%)	17 (30.9%)	23 (28.8%)	21 (26.6%)	3 (5.3%)	12 (16.0%)
	Bachelor degree	10 (2.4%)	0 (0%)	0 (0%)	7 (8.8%)	0 (0%)	0 (0%)	3 (4.0 %)
3	Occupation							
	Agriculture	266 (62.6%)	56 (70.9%)	51 (92.7%)	45 (56.3%)	12 (15.2%)	53 (93.0%)	49 (65.3%)
	Undergraduate student	2 (0.5%)	1 (1.3%)	0 (0%)	0 (0%)	0 (0%)	0 (0%)	1 (1.3%)
	Government officer	4 (0.9%)	1 (1.3%)	1 (1.8%)	1 (1.3%)	0 (0%)	0 (0%)	1 (1.3%)
	Private business	2 (0.5%)	0 (0%)	0 (0%)	2 (2.5%)	0 (0%)	0 (0%)	0 (0%)
	Housewife	11 (2.6%)	0 (0%)	0 (0%)	5 (6.3%)	2 (2.5%)	0 (0%)	4 (5.3%)
	Freelance	110 (25.9%)	21 (26.6%)	1 (1.8%)	19 (23.8%)	57 (72.2%)	1 (1.8%)	11 (14.7%)
	Merchant	8 (1.9%)	0 (0%)	1 (1.8%)	2 (2.5%)	2 (2.5%)	0 (0%)	3 (4.0%)
	Private company	2 (0.5%)	0 (0%)	1 (1.8%)	0 (0%)	1 (1.3%)	0 (0%)	0 (0%)
	Others	20 (4.7%)	0 (0%)	0 (0%)	6 (7.5%)	5 (6.3%)	3 (5.3%)	6 (8.0%)
4	Income (baht/month)							
	<5000	253 (59.5%)	53 (67.1%)	36 (65.5%)	40 (50.0%)	30 (38.0%)	51 (89.5%)	43 (57.3%)
	5001–10,000	125 (29.4%)	19 (24.1%)	15 (27.3%)	31 (38.8%)	36 (45.6%)	6 (10.5%)	18 (24.0%)
	10,001–15,000	24 (5.7%)	3 (3.8%)	1 (1.8%)	7 (8.8%)	11 (13.9%)	0 (0%)	2 (2.7%)
	15,001–20,000	4 (0.9%)	1 (1.3%)	1 (1.8%)	0 (0%)	2 (2.5%)	0 (0%)	0 (0%)
	20,001–25,000	6 (1.4%)	2 (2.5%)	1 (1.8%)	1 (1.3%)	0 (0%)	0 (0%)	2 (2.7%)
	>25,000	13 (3.6%)	1 (1.3%)	1 (1.8%)	1 (1.3%)	0 (0%)	0 (0%)	10 (13.3%)

**Table 2 ijerph-17-01698-t002:** Influencing factors in selection of drinking water.

Factors	Tribes (Max = 5)												*F*-Test *p*-Value
	Akha (*n* = 79)		Hmong (*n* = 55)		Karen (*n* = 80)		Lahu (*n* = 79)		Lisu (*n* = 57)		Yao (*n* = 75)		
	Mean (SD)		Mean (SD)		Mean (SD)		Mean (SD)		Mean (SD)		Mean (SD)		
Colorless or clear	4.0	(0.9)	4.6	(0.6)	4.7	(0.7)	4.7	(0.6)	3.9	(0.5)	4.7	(0.6)	* 0.000
No taste	3.4	(1.2)	3.7	(1.1)	4.3	(1.1)	4.0	(1.2)	4.8	(0.4)	3.4	(1.2)	* 0.000
No odor	3.6	(1.0)	4.2	(0.9)	4.7	(0.7)	4.6	(0.9)	4.7	(0.7)	4.9	(3.2)	* 0.000
Low cost	3.9	(1.0)	4.0	(1.0)	4.6	(0.9)	4.4	(0.9)	4.0	(0.3)	2.5	(1.3)	* 0.000
Disinfection process	3.4	(1.0)	3.9	(1.2)	4.4	(0.9)	4.4	(1.0)	4.0	(0.3)	4.1	(1.0)	* 0.000
Easy access	3.6	(0.9)	3.8	(1.1)	4.4	(0.9)	4.1	(1.1)	4.8	(0.4)	3.3	(1.0)	* 0.000
Sufficiency	3.6	(0.8)	3.5	(1.0)	4.2	(1.1)	4.5	(0.8)	4.9	(0.4)	3.4	(1.0)	* 0.000
Health safety	3.9	(0.9)	4.2	(0.9)	4.7	(0.7)	4.8	(0.5)	4.9	(0.3)	3.9	(1.0)	* 0.000
FDA standard ^a^	3.1	(1.1)	3.7	(1.2)	4.6	(0.8)	4.7	(0.8)	3.1	(0.3)	3.9	(0.9)	* 0.000
Advertisement	2.8	(1.3)	3.0	(1.2)	3.9	(1.1)	4.1	(1.1)	3.1	(0.3)	2.4	(0.9)	* 0.000

* a significant level (*p* < 0.05). ^a^ FDA standard is defined only the labeled sign of the Food and Drug Administration Certification of Ministry of Public Health, Thailand.

**Table 3 ijerph-17-01698-t003:** Influencing factors of drinking water selection from three sources of drinking water.

Factors	Drinking Water Sources						*F*-Test *p*-Value
	Mountain Water Supply (*n* = 316)		Bottled Drinking Water (*n* = 91)		Shallow Wells (*n* = 14)		
	Mean (SD)		Mean (SD)		Mean (SD)		
Colorless or clear	4.5	(0.8)	4.5	(0.7)	4.4	(0.5)	0.986
No taste	3.9	(1.2)	3.8	(1.1)	4.4	(0.9)	0.199
No odor	4.4	(1.0)	4.6	(2.9)	4.4	(0.8)	0.368
Low cost	3.9	(1.2)	3.7	(1.1)	3.8	(0.9)	0.368
Disinfection process	4.1	(1.1)	3.9	(0.9)	4.1	(0.8)	0.360
Easy access	4.0	(1.1)	3.7	(1.1)	4.5	(0.5)	* 0.006
Sufficiency	4.1	(1.0)	3.6	(1.1)	4.4	(0.7)	* 0.000
Health safety	4.4	(0.9)	4.3	(0.8)	4.5	(0.9)	0.428
FDA standard	3.8	(1.2)	4.3	(0.9)	3.9	(0.8)	* 0.001
Advertisement	3.2	(1.2)	3.3	(1.1)	3.4	(0.6)	0.562

* a significant difference (*p* < 0.05).

**Table 4 ijerph-17-01698-t004:** Mountain water supply management of six hill tribe groups, Chiang Rai, Thailand.

Tribe	Location	Altitude (MASL)	Sources of Raw Water	Water Reservoir/Storage	Type of Pipe	Maintenance Cost (US Dollars/Month)	Problems Identified by Interviews
Akha	N 20°19′33.06″ E 100°18′45.04″–N 20°18′10.12″ E 099°39′27.79″	961–977	4 points of raw water source	4 concrete dams, 3 concrete storage tanks	PVC, Steel	124	(1) Solid particle accumulated at bottom of the dams and tanks (2) PVC pipe damages (3) Raw water polluted from nearby community
Hmong	N 20°02′43.74″ E 099°53′38.27″–N 20°02′43.74″ E 099°53′38.27″	700–810	2 points of raw water source	2 concrete dams, 1 concrete storage tank	PVC	7.9	(1) PVC pipe damages (2) Turbidity increased during raining.
Karen	N 19°57′35.86′’ E 099°42′46.50″–N 19°57′36.13′’ E 099°42′40.27″	405–451	2 points of raw water source	2 concrete dams, 1 concrete storage tank	PVC	39.5	(1) PVC pipe damages (2) Inadequate raw water during dry season
Lahu	N 20°03.142′ E 099°49.458′–N 20°02.936′ E 099°49.336′	471–571	2 points of raw water source	2 concrete dams, 1 concrete storage tank	PVC	25	(1) Solid particle accumulated at bottom of the dams and tanks (2) PVC pipe damages
Lisu	N 20°21′55.48″ E 099°28′11.50″–N 20°22′03.56″ E 099°28′19.46″	1140–1232	1 points of raw water source	1 concrete dam, 1 concrete storage tank	PVC	8	(1) Solid particle accumulated at bottom of the dams and tanks (2) PVC pipe damages (3) Inadequate raw water during dry season
Yao	20°19′33.06″ E 100°18′45.04″–N 20°19′33.06″ E 100°18′45.04″	540–634	7 points of raw water source	3 concrete dams, 4 soil dams 2 concrete storage tanks	PVC, Steel	151	(1) Solid particle accumulated at bottom of the dams and tanks (2) PVC pipe damage

**Table 5 ijerph-17-01698-t005:** Water quality of mountain water supply of six hill tribe groups.

Tribe	Number of Water Sampling (*n*)	Parameters		
		Physical Parameter	Biological Parameter	
		Turbidity (Unit: NTU)	Total Coliform Bacteria (Unit: 100 mL MPN)	Fecal Coliform Bacteria (Unit: 100 mL MPN)
		Min–Max	Min–Max	Min–Max
Akha	4	2.10–8.97	33–130	7–49
Hmong	4	0.64–2.83	<2–540	<2–540
Karen	4	1.09–2.62	23–33	23–33
Lahu	4	5.14–9.39	33–79	33–79
Lisu	4	0.74–10.23	<2–920	<2–920
Yao	4	0.77–12.53	5–130	2–49
Total	24	0.64–12.53	<2–920	<2–920
Drinking Water Standard		Less than 5 [12,32]	Must not be detected [12,32]	Must not be detected [12]
Ratio: Not meet standard/Total sample (%)		10/24 (41.67%)	24/24 (100%)	24/24 (100%)

[12] WHO, 2011, Standard of drinking water. [32] Department of Health, 2010, Standard of drinking water from water supply system.

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
