# Peer review of "Drinking Water Investigation of Hill Tribes: A Case Study in Northern Thailand"

_ijerph, 2020, doi:10.3390/ijerph17051698_

Round 1
Reviewer 1 Report
This paper has made a detailed investigation on the drinking water of the tribes in the northern mountainous areas of Thailand, which is very meaningful for mastering the local drinking water situation and giving guidance, but the paper needs some modifications before accepting.
The English needs further revision. There are lots of “it found”, “it indicated”… should change to “it was founded that”, “it was indicated that”… The main work of this paper is to investigate the drinking water situation. The data given in the article is detailed, but there is not enough discussion and management guidance. Therefore, it is necessary to strengthen the discussion. And I suggest that the title of the paper can be changed to "Drinking water investigation of hill tribes: case study in northern Thailand "Author Response
Response to Reviewer 1 Comments
Point 1: The English needs further revision. There are lots of “it found”, “it indicated”… should change to “it was founded that”, “it was indicated that”… The main work of this paper is to investigate the drinking water situation. 

Response 1: We change following the reviewer comment.
Point 2: The data given in the article is detailed, but there is not enough discussion and management guidance.
Response 2: We write more discussion and management guidance in our paper.
1) This similar to previous studies that found the mountain water supply was a primary source of freshwater supplied in a hilly region including the hill tribe people on a mountainous area of Thailand [9, 11, 21]. Bottled drinking water was also a drinking water source of Yao, Karen, Akha, and Lahu, but not for Hmong and Lisu. Some people believe that a bottled drinking water is convenient and has better taste than tap water, so its consumption is increasing worldwide [22]. However, Hmong and Lisu did not prefer bottled drinking water. It might be due to transport difficulty, since their villages are located far away from the urban zone. Besides, it was found that some uses of the shallow wells water by Lisu, Hmong, Karen of about 12.28%, 7.50%, and 1.33%, respectively. The low consumption of shallow wells water, especially Hmong and Karen, might be due to accessibility because their villages are closed to the large river.
2) Water with turbidity value less than 5 NTU is defined as a clear water, and a person can begin to see cloudiness with the naked eyes when turbidity value higher than 5 NTU [12, 23-24].
3) Odor can originate from algae or microorganisms in a water source, resulting from growing, activities or metabolism [25]. Drinking water with acceptable in appearance, taste, and odor is a high priority for consumer [12].
4) Due to the hill tribe people live in the mountainous area, mountain water supply is a primary source of freshwater [26]. Whereas, bottled drinking water was selected with low percentage due to difficult to access. However, the results showed some hill tribe groups, including Yao, Karen, Akha, and Luha, consuming bottled drinking water of about 45.33%, 33.75%, 24.05%, and 13.92%, respectively. These groups concerned more about the FDA standards guarantee for drinking water. Besides, the results from the survey question about knowledge of safe drinking water characteristics were found that the mean score of the knowledge of Yao was significantly higher than the other groups (p < 0.05), suggesting drinking water selection might be related to knowledge.
5) While all water samples of Lahu were over the standards, which might be caused by a too small storage tank, leading to high water and particle turbulence affected suspended particle sedimentation [33]. To improve sedimentation efficiency, the size of storage tank should be designed based on water velocity and water volume [33]
6) Besides, especially in the raining season, small particles, sediments, suspended solids, bacteria, and organic substances from storm water run-off were a major cause of high turbidity of the raw water [21].
7) Therefore, a simple disinfection method such as boiling which can inactivate of about 99% of geometric mean fecal coliforms [37], is recommended.
Point 3: I suggest that the title of the paper can be changed to "Drinking water investigation of hill tribes: case study in northern Thailand.
Response 3: We changed to “Drinking water investigation of hill tribes: case study in northern Thailand”
Reviewer 2 Report
Drinking water management of hill tribes: case study in northern Thailand
Thank you for the opportunity to review this paper.
To begin with, I applaud the authors for their hard work and efforts in extensively gathering the data to understand this salient topic. This type of publication would seem well suited to International Journal of Environmental Research. However, before publication in this journal or any other scientific journal authors should address ALL or MOST of my concerns.
One of the salient shortcomings of this paper is that it is very descriptive without any specific hypothesis. Secondly the manuscript is too long – the authors use more words than necessary thus drifting too far afield in reporting everything. The method and results sections require a bit of reorganization, at the moment there is lack of coherence and flow. Some of the text in the result section should be moved to the introduction/methods or removed. The ‘éclat’ of this paper is utterly buried under the rubble.
This manuscript has many typos and numerous linguistic errors (e.g. verb agreement) that at times make it difficult to follow. I would suggest that it may be useful to engage a professional English language editor following a restructure of the paper. These must be addressed.
Specific comments:
- I find the title a little too general. Maybe replace management with practices. Edit.
- Delete ‘water is essential to human life.’
- L78-80. “The 6 hill tribe villages are located at altitudes of 410 - 1,300 MASL and latitude-longitude of 78 N19o57’35.86’’ E099o42’46.50” - N20o21’55.48” E099o28’11.50”. All study sites were determined the 79 location and elevation by using global position system (GPS) (Garmin Oregon 750, U.SA.)”. Redundant. Delete.
- The figures and tables could use minor adjustments to improve legibility.
- For the tables and throughout the manuscript, round your decimals to one degree, i.e 2.6 and not 2.59) to reduce clutter.
- Maybe explore use of alternate shading in the tables to guide the reader.
- I’m not conversant with Item-objective congruence. If IOC is not a measure of reliability, could the authors comment on the reliability (using either Cronbach’s alpha or item response theory) of their test questions in answering the underlying constructs?
- Line 14: Drinking water mainly obtains from untreated natural water sources that can cause health problems. Replace with: Drinking water is mainly obtained from untreated water sources that can cause health problems
- Line 79 – 80: Unclear. Rewrite
- Line 87: The cited article (19) does not discuss Item objective congruence: Replace with relevant reference.
- Line 116-121: Repetition, re-write.
- What microbiological tests were done?
- Line 126: Explain what other sources are available
- Line 149-153: Move to methods section
- Line 152: What type of Multiple Comparison analysis was used. This should be explained in the methods section
- Line 176: The table shows p value of 0.00 for all the factors, why is this so?
- The use of Likert scale is first mentioned in results section! Should have been explained in the methods section
- The criterion for selection of the study areas (the six tribes) needs to be clearly explained in the methods section
- Ensure all the methods used in this study are systematically explained in the methods section
- The factors influencing drinking water selection ought to have been explained in the methods section. Currently it is in the results and discussion section
- The criterion used to select how households were selected for questionnaire administration needs to be explained clearly as well as sample size calculation if any was used
- How the sources of water to be tested was selected needs to be explained
- Replace pie charts with bar graphs. The former is hard to make comparisons across.
- Whilst it’s ok to combine results and discussion sections, as done here, authors need to discuss their findings clearly where needed. As is, most of the findings are not discussed. For example, how do your findings compare with other studies carried out in similar settings In Thailand/Asia/Africa etc.
- How would the participants know about the FDA standards?
- Authors’ choice of statistics analytical methods is unclear. In the first instance they use One-Way Anova/T test– both parametric methods assuming a normal distribution of data – yet on the other hand they explore the use of Kruskal Wallis – a non- parametic technique. Whilst it’s ok to use different methods, authors should clearly explain their rationales. Did you test the data for normality? Did you correct for multiple testing i.e. using Bonferroni or FDR ?
- Statistical analysis: Was the distribution of data normal? Why apply both parametric and non-parametric test i.e. One-way ANOVA and Kruskal Wallis for the same data set?
- L151-153 Clarification. Did you create a composite index out of the likert scale by adding up the values? If so, I think this is not the right approach as Likert scale measures an underlying construct which is just not an obvious additive process. Comment.
- L170-175. I find your presentation style using greater than sign > very odd. Write in prose.
- Table 2. I find the all significant p values result very suspicious. Kindly check.
- What methods was used for post hoc testing?
- Is it necessary to correct for FDA standard given that these standards are not global but based on American water requirements. Are there local/regional standards? Similarly, I find it perplexing that hill tribes (L198 and L275) were concerned about FDA standards? Is this an artefact (questionnaire administration bias) as opposed to an actual result?
- Indicate the monetary value in US dollars in brackets for global audience.
- L206-225. Summarize to just a few lines. Too wordy!
- How important is table 4. Maybe move to supplementary section.
- The finding that the majority of hill tribes boiled their drinking water and that a small minority did not but instead used water filters is remarkable and an important intervention point.
- I’m intrigued to know whether the hill tribes knew or were concerned with water safety. This could explain why most of them drank untreated water. Comment.
Author Response
Response to Reviewer 2 Comments
Point 1: I find the title a little too general. Maybe replace management with practices. Edit.
Response 1: We changed to “Drinking water investigation of hill tribes: case study in northern Thailand”
Point 2: Delete ‘water is essential to human life.’
Response 2: Delete ‘water is essential to human life’ following reviewer comments
Point 3: L78-80. “The 6 hill tribe villages are located at altitudes of 410 - 1,300 MASL and latitude-longitude of 78 N19o57’35.86’’ E099o42’46.50” - N20o21’55.48” E099o28’11.50”. All study sites were determined the 79 location and elevation by using global position system (GPS) (Garmin Oregon 750, U.SA.)”. Redundant. Delete.
Response 3: We edited to all study sites were determined the 72 locations and elevation by using global position system (GPS) (Garmin Oregon 750, U.SA.)”.
Point 4: The figures and tables could use minor adjustments to improve legibility.
Response 4: We improved the figure and tables.
Point 5: For the tables and throughout the manuscript, round your decimals to one degree, i.e 2.6 and not 2.59) to reduce clutter.
Response 5: We changed the decimals to one degree for all tables.
Point 6: Maybe explore use of alternate shading in the tables to guide the reader.
Response 6: We explored use of alternate shading by add Bold and Italic numbers
Point 7: I’m not conversant with Item-objective congruence. If IOC is not a measure of reliability, could the authors comment on the reliability (using either Cronbach’s alpha or item response theory) of their test questions in answering the underlying constructs?
Response 7: Please provide your response for Point 2. (in red)
This questionnaire was tied out for the reliability analysis which 20% of sample size was carried out for the questionnaire tried out . The Cronbach’s alpha was then analyzed to confirm the reliability of this questionnaire. From this analysis, it was found that the Cronbach’s alpha of this questionnaire was 0.087 that was the high level of the reliability.
Point 8: Line 14: Drinking water mainly obtains from untreated natural water sources that can cause health problems. Replace with: Drinking water is mainly obtained from untreated water sources that can cause health problems
Response 8: We replaced “Drinking water is mainly obtained from untreated water sources that can cause health problems”.
Point 9: Line 79 – 80: Unclear. Rewrite
Response 9: We rewrite to All study sites were determined the 72 locations and elevation by using global position system (GPS) (Garmin Oregon 750, U.SA.)”.
Point 10: Line 87: The cited article (19) does not discuss Item objective congruence: Replace with relevant reference.
Response 10: We cited the article ;Turner RC.; & Carlson L. Indexes of Item-Objective Congruence for Multidimensional Items. Journal International Journal of Testing. 2003,3,2. https://doi.org/10.1207/S15327574IJT0302_5.
Point 11: Line 116-121: Repetition, re-write.
Response 11: We cut the repetition sentences.
Point 12: What microbiological tests were done?
Response 12: The microbiological tests were done following the standard methods of the 9221 B and 9221 E.
Point 13: Line 126: Explain what other sources are available
Response 13: Other sources are available such as rain water, spring, and small pond.
Point 14: Line 149-153: Move to methods section
Response 14: We move to methods section
Point 15: Line 152: What type of Multiple Comparison analysis was used. This should be explained in the methods section
Response 15: We use only One-way ANOVA.
Point 16: Line 176: The table shows p value of 0.00 for all the factors, why is this so?
Response 16: We confirm that the table shows p value of 0.00 for all the factors
Point 17: The use of Likert scale is first mentioned in results section! Should have been explained in the methods section
Response 17: We explained Likert scale in methods section
Point 18: The criterion for selection of the study areas (the six tribes) needs to be clearly explained in the methods section
Response 18: We choose only major hill tribes from 17 hill tribes in Chiang Rai which more number of populations.
Point 19: Ensure all the methods used in this study are systematically explained in the methods section
Response 19: Yes
Point 20: The factors influencing drinking water selection ought to have been explained in the methods section. Currently it is in the results and discussion section
Response 20: The factors influencing drinking water selection reviewed from previous studies and characteristic of water standard.
Point 21: The criterion used to select how households were selected for questionnaire administration needs to be explained clearly as well as sample size calculation if any was used
Response 21: The criterion used to select how households were selected for questionnaire. We survey total household in each hill tribe and random household greater than 40% of each hill tribe that respective of water quality system.
Point 22: How the sources of water to be tested was selected needs to be explained
Response 22: In addition, water samples were collected by grab sampling from (1) water sources on the mountain, (2) water storage tanks, and (3) water at households, between March – April 201
Point 23: Replace pie charts with bar graphs. The former is hard to make comparisons across.
Response 23: The purpose of our study no needs to compare between each hill tribes. We think pie chart it’s very clear.
Point 24: Whilst it’s ok to combine results and discussion sections, as done here, authors need to discuss their findings clearly where needed. As is, most of the findings are not discussed. For example, how do your findings compare with other studies carried out in similar settings In Thailand/Asia/Africa etc.
Response 24: We added new findings compare with other studies carried out in similar settings in hill tribe, Thailand and Asia.
Point 25: How would the participants know about the FDA standards?
Response 25: FDA standard is the Food and Drug Administration certification of Ministry of Thailand which Thai government promote with people.
Point 26: Authors’ choice of statistics analytical methods is unclear. In the first instance they use One-Way Anova/T test– both parametric methods assuming a normal distribution of data – yet on the other hand they explore the use of Kruskal Wallis – a non- parametic technique. Whilst it’s ok to use different methods, authors should clearly explain their rationales. Did you test the data for normality? Did you correct for multiple testing i.e. using Bonferroni or FDR ?
Response 26: Normality tests were used Kolmogorov–Smirnov test. One-way ANOVA was applied to compare the mean scores among sample groups, whereas t-test for parametric methods and Kruskal-Wallis test were applied to compare the mean scores among sample groups for non- parametric methods. The significant difference was considered if the p-value was less than 0.05.
Point 27: Statistical analysis: Was the distribution of data normal? Why apply both parametric and non-parametric test i.e. One-way ANOVA and Kruskal Wallis for the same data set
Response 27: Normality tests were used Kolmogorov–Smirnov test. One-way ANOVA was applied to compare the mean scores among sample groups, whereas t-test for parametric methods and Kruskal-Wallis test were applied to compare the mean scores among sample groups for non- parametric methods. The significant difference was considered if the p-value was less than 0.05.
Point 28: L151-153 Clarification. Did you create a composite index out of the likert scale by adding up the values? If so, I think this is not the right approach as Likert scale measures an underlying construct which is just not an obvious additive process. Comment.
Response 28: We explained Likert scale in methods section
Point 29: L170-175. I find your presentation style using greater than sign > very odd. Write in prose.
Response 29: We changed presentation style using greater than sign.
Point 30: Table 2. I find the all significant p values result very suspicious. Kindly check.
Response 30: We confirm that the table shows p value of 0.00 for all the factors
Point 31: What methods was used for post hoc testing?
Response 31: LSD for homogeneity of equal variance and Dunnett’s T3 for unequal variance.
Point 32: Is it necessary to correct for FDA standard given that these standards are not global but based on American water requirements. Are there local/regional standards? Similarly, I find it perplexing that hill tribes (L198 and L275) were concerned about FDA standards? Is this an artefact (questionnaire administration bias) as opposed to an actual result?
Response 32: FDA standard is the Food and Drug Administration certification of Ministry of Thailand which Thai government promote with people (local standard).
Point 33: Indicate the monetary value in US dollars in brackets for global audience
Response 33: We indicated the monetary value in US dollars in brackets for global audience
Point 34: L206-225. Summarize to just a few lines. Too wordy!
Response 34: We change to “The mountain water supply management of the 6 hill tribes is summarized in Table 4 and Figure 4. Mountain stream water is obtained from 1-7 points of raw water source depending on the topography of each hill tribe located. Concrete dams, soil dams and water storage concrete-tanks constructed to collect the mountain stream water were far from the villages about 1-6 km and close to the Thai - Myanmar border (Akha and Lisu). The collected water was supplied through 2-inch and 3-inch PVC pipes to water storage tanks installed at each hill tribe village. The water was stored in these tanks to allow suspended particles to settle out of the water before further distribute to each household using 1½ inch and ½ inch PVC pipes. This mountain water supply system is commonly applied in hilly regions [21]”.
Point 35: How important is table 4. Maybe move to supplementary section.
Response 35: The table can help the reader to more understanding about mountain water supply management.
Point 36: The finding that the majority of hill tribes boiled their drinking water and that a small minority did not but instead use water filters is remarkable and an important intervention point.
Response 36: Boiled water is cheaper process more than filter machine installation, so boiled become a popular process.
Point 37: I’m intrigued to know whether the hill tribes knew or were concerned with water safety. This could explain why most of them drank untreated water. Comment
Response 37: Because they live in remote area cannot accessed the equipment to clean water that is limitation of water treatment.
Reviewer 3 Report
Good job by the authors in conducting a water access survey in remote parts of Northern Thailand. The data provides new observations to that region and its residents.
As for suggestions for improvements, there are a few. First, there grammatical errors scattered in the paper. They are frequent enough to warrant a thorough review to flush them out.
Second, the research is purely descriptive. It does not provide an explanation to why your team is seeing that particular data emerging from your survey. This is problematic, as without an explanation and its thorough exploration, it is unclear whether your recommendations would even be endorsed. Especially, more so, when your recommendation is voluntary compliance and will require the commitment of additional resources or labor for fuel.
This is unfortunate since in the beginning of your paper, you noted that these are villages with their own cultures and beliefs. However, you do not mention and discuss how these cultures and beliefs may have a factor in the particular survey outcomes. This would have may this paper much more significant since it would open a discussion on the resiliency of practices and beliefs which are sustained by their cultures and beliefs. Given that all water has coliform bacteria contamination, I am wondering why you are not reporting or collected data on water-born disease outbreak rates. This information should have been collected and analyzed. This is because it would address a much more important question of whether certain villagers are stuck to certain ways of acquiring and consuming water despite suffering from water-borne diseases.
Given that the merit of this lies in sharing collected data, I question what is significantly new that warrants scholars in the field to pay attention to this. Descriptions about the access levels and their alternatives sources are not something entirely new. That is, it is what is expected. If I had to dig deeper in the relevance of your work, it lies in the perception rankings. However, your team does not expand on this. That is, the perception ranking is presented but not explained. This needs to be strengthened. I say this because even the water management system description is not something new. It is a typical description of a small scale water system. Thus, it leads me to wonder what is so different that offers new insight to the field.
This leads to my conclusion. I do appreciate the survey data and its presentation. However, I do recommend that the researchers revisit and revise their paper to better highlight the significance of their paper. That is, the researchers need to answer, "what additional insights are generated from this study that pushes the scholarship forward?"
Author Response
Response to Reviewer 3 Comments
Point 1: First, there grammatical errors scattered in the paper. They are frequent enough to warrant a thorough review to flush them out.
Response 1: We checked grammatical errors scattered in the paper.
Point 2: Second, the research is purely descriptive. It does not provide an explanation to why your team is seeing that particular data emerging from your survey. This is problematic, as without an explanation and its thorough exploration, it is unclear whether your recommendations would even be endorsed. Especially, more so, when your recommendation is voluntary compliance and will require the commitment of additional resources or labor for fuel.
Response 2: We write more discussion and management guidance in our paper.
Water treatment suggestion
The research team has compiled relevant recommendations for efficient drinking water management in 3 levels, which are household level, community level (village), and policy level.
Household-level: Recommendations for hill tribe s’ household in short-term drinking water improvement from the mountain water supply. We recommend to; (1) improved the turbidity by filtering methods or using primary alum then boiling to kill germs before consumption. (2) Install household water filters with disinfection systems such as UV light, ozone, chlorine or reverse osmosis systems, etc. The water filter system in the household will find clogging problems from suspended solids in the rainy season with high turbidity values. Hill tribe people should have a plan for regularly changing filter columns. (3) Promote knowledge and skills for efficient drinking water management for hill tribe people in each household.
Community-level (village): (1) Set a network of consumer water supply for mountain water and makes the agreements and regulations about the mountain water supply. (2) Conserve and maintain the dam areas as a conservation area for preventing contamination of mountain water supply, such as prohibiting agriculture, animal feeding, building, and deforestation. For example, Hmong tribes have designated conservation areas called "community forests", establishing rules to prevent encroachment of forest areas. (3) Promote community leaders and mountain water system administrators to have professional water supply management skills. (4) Create sediment and turbidity limit systems. Alum is applying help precipitate of suspended solids and sand filtration in the community. (5) Install disinfection systems such as chlorine-filling systems, UV systems, ozone systems, or reverse osmosis systems. (6) Check drinking water quality in the community at least one time per month.
Policy level: The local government organization supports, promotes, and trains skills, and support budgets to manage the mountain water supply system appropriately.
Point 3: This is unfortunate since in the beginning of your paper, you noted that these are villages with their own cultures and beliefs. However, you do not mention and discuss how these cultures and beliefs may have a factor in the particular survey outcomes.
Response 3: All hill tribe was not more previous study about water management in each hill tribe. We added only about Akha following “Meanwhile, most of Akha and Lahu have stored the water in containers to allow particle settling before drinking without other treatments. These results were similar to the previous reports [9, 29]. It might be due to their belief that the boiled water is given only for an ill person [30].
Point 4: Given that all water has coliform bacteria contamination, I am wondering why you are not reporting or collected data on water-born disease outbreak rates.
Response 4: Just report in next research and during finding the data.
Point 5: Given that the merit of this lies in sharing collected data, I question what is significantly new that warrants scholars in the field to pay attention to this.
Response 5: The significantly new that warrants scholars in the field to pay attention to this was the hill tribes are group of people who live in high mountains, remote and difficult to access areas. Most of water supply and drinking water were obtained from untreated natural water sources such as mountain water, groundwater and shallow wells. Next research, we will set standard procedure for mountain drinking water for hill tribe area.
Round 2
Reviewer 1 Report
this artical could be acceptted now.
Author Response
Point 1: English language and style are fine/minor spell check required
Respond : Our researchers hardly improved grammar errors and spelling check of whole manuscript following attachment file.

Reviewer 2 Report
Thanks for working on my comments.
Author Response
Point 1: Moderate English changes required
Respond : Our researchers hardly improved grammar errors and spelling check of whole manuscript following attachment file.

Reviewer 3 Report
This manuscript requires extensive revision to flush out various grammatical errors. As demonstration, I have listed below the various grammatical problems just in the Introduction. (The remaining document is also filled with grammatical errors - it makes no sense for me to point all of it out.) I suggest that the authors get a native speaker to review and correct all the grammatical errors.
The first highlighted yellow should be "found that" instead of "were found that."
3rd sentence into the introduction needs to be rewritten. "In 2017, about 5.3 billion people accessed to safely managed drinking-water services, but the remaining 2.2 billion people could not access to the services" needs to be changed to "In 2017, about 5.3 billion people had access to safely managed drinking drinking-water services, but the remaining 2.2 billion people still do not have that access."
Highlighted 6th line in the Introduction needs to change from 'WHO was found that" to "WHO found."
Line 10 in Introduction: "contributes" should be "is responsible for"
Line 11 in Introduction: "may originate by" should be "originates from"
1st line on page 2: "defecation closed to" should be "defecation close to"
8th line on page 2: "eventuate a benefit" should be "eventually benefit"
11th line on page 2: "The general water treatment system begins with raw water receiving follows by coagulation and flocculation, sand filtration, disinfection, and water distribution to the consumers" should be "The general water treatment system begins with raw water undergoing coagulation and flocculation, sand filtration, and disinfection. After these processes, the treated water is then distributed to the consumers."
3rd paragraph on page 2: "in the Northern Thailand" needs to be "in Northern Thailand"
3rd line in the 3rd paragraph of page 2: Change to "They have migrated from the southern part of China to Thailand in recent decades."
4th line in the 3rd paragraph of page 2: "hill tripe" should be "hill tribe"
6th line in the 3rd paragraph of page 2: "In year 2012, there were 180,214 hill tribe people live in 652 villages in Chiang Rai province" should be "In 2012, 180,214 hill tribe members spread over 652 villages resided in the Chiang Rai province."
8th line in the 3rd paragraph of page 2: "Their villages locate" should be "Their villages are located"
9th line in the 3rd paragraph of page 2: "the 6 hill tribes in this study areas" should be "the 6 hill tribe areas in this study"
last sentence in the 3rd paragraph of page 2 needs to be rewritten for clarity. "This research aimed' should be "This research aims." The "regarding their knowledge, culture, and environment" needs further explanation since it does not mesh well with the sentence.
Author Response
Point 1: Extensive editing of English language and style required
Response : We changed all comments following reviewer 3 as show yellow highlight in attachment file.
